# Association between Coffee Consumption, Caffeine Intake, and Metabolic Syndrome Severity in Patients with Self-Reported Rheumatoid Arthritis: National Health and Nutrition Examination Survey 2003–2018

**DOI:** 10.3390/nu15010107

**Published:** 2022-12-26

**Authors:** Shuolin Wang, Yingdong Han, He Zhao, Xinxin Han, Yue Yin, Juan Wu, Yun Zhang, Xuejun Zeng

**Affiliations:** Department of Family Medicine & Division of General Internal Medicine, Department of Medicine, Peking Union Medical College Hospital, Chinese Academy of Medical Sciences, State Key Laboratory of Complex Severe and Rare Diseases (Peking Union Medical College Hospital), Beijing 100730, China

**Keywords:** coffee, caffeine, rheumatoid arthritis, metabolic syndrome, National Health and Nutrition Examination Survey

## Abstract

Rheumatoid arthritis (RA) is chronic inflammatory disease. Although coffee impacts metabolism, no evidence has shown an association between coffee consumption and decreased risk for developing metabolic syndrome (MetS) among RA patients. Hence, we examined the association between coffee consumption and metabolic syndrome severity among 1094 participants with self-reported RA. Accordingly, patients with MetS z-scores of <0 and ≥0 were designated as low- and high-risk groups, respectively. In the fully adjusted model, drinking over two cups of coffee daily was associated with a decrease in the MetS z-score (*p* = 0.04). Subgroup analysis showed that in the low-risk group, daily intake of <2 cups of coffee was associated with low MetS z-scores (*p* = 0.003), scores (*p* = 0.03). Coffee intake was associated with low body mass index (*p* = 0.03 for 0–2 cups per day; *p* = 0.02 for >2 cups per day) and low HOMA-IR (β, −2.62; 95%CI, −5.13 to −0.11; *p* = 0.04). Our study suggests that coffee, but not decaffeinated coffee consumption and total caffeine intake, is associated with MetS severity in RA.

## 1. Introduction

Rheumatoid arthritis (RA) is a common chronic inflammatory condition that primarily affects the joints [1]. In Europe and North America, its prevalence ranges from 0.5% to 1.0%, with some Native American populations showing high prevalence [2]. Recent evidence suggests that RA contributes significantly to the growing health care costs in the United States [3]. While the exact mechanism by which RA develops and progresses is unknown, inflammation has always been considered a key factor throughout the course of the disease [1]. Currently, the treatments available for RA, including nonsteroidal anti-inflammatory drugs, disease-modifying antirheumatic drugs, and glucocorticoids, have diversified and prolonged survival. However, despite the remarkable progress in therapeutics, partial or non-responders have still been observed, and the incidence of chronic co-morbidities, including metabolic syndrome (MetS) and cardiovascular disease, has been increasing over time [1,4]. Therefore, the recent recommendations of the European League Against Rheumatism (EULAR) for the management of these significant medical challenges include lifestyle interventions via a multidimensional approach [5].

RA has been associated with a higher incidence of MetS and insulin resistance due to the hyperinflammatory state, which poses a challenge for treatment. MetS refers to a cluster of factors, such as central obesity, hypertension, high triglycerides, and low high-density lipoprotein cholesterol levels (HDL-C). The underlying pathophysiology of MetS is thought to be associated with insulin resistance [6]. Evidence in patients with RA has shown that the disease state and MetS influence each other. On the one hand, inflammation associated with RA itself and the drugs used to treat it, including glucocorticoids, can lead to metabolic abnormalities. On the other hand, the abnormal control of glucometabolic and lipid metabolism promotes the development of inflammation. The complex interactions between inflammation, treatment, and metabolism increase the prevalence and severity of MetS in patients with RA to some extent. However, the presence of MetS can still partly limit the treatment of RA. Therefore, nutritional intake has gained widespread attention from patients and physicians [7]. Estimates have shown that 500 billion cups of coffee are consumed annually around the world, making it one of the most popular beverages worldwide [8]. Caffeine may reduce the risk of developing type 2 diabetes mellitus (T2DM), one of the key components of MetS [8]. However, current research remains controversial regarding the effects of coffee consumption on MetS due to different study designs and study populations. A case–control study including 250 people suggested two to three cups of coffee a day was inversely associated with the incidence of MetS [9]. Another cross-section survey from the HAPIEE study that included 8821 adults showed that coffee was associated with reduced risk of MetS [10]. However, a population-based prospective cohort study by Lutsey et al. that included 9514 middle-aged adults (45 to 64 years old) and examined the relationship between their dietary data and MetS ultimately found no relationship between the incidence of MetS and coffee consumption [11]. Similarly, an epidemiological study based on a Chinese cohort did not suggest a significant association between MetS and coffee intake [12]. Moreover, no evidence has shown an association between coffee consumption and a lower likelihood of developing MetS among RA patients in the complex context of inflammation and metabolism.

The current study examined the association between coffee consumption and MetS severity among 1094 participants in the National Health and Nutrition Examination Survey (NHANES) 2003–2018 database. To the best of our knowledge, this has been the first study to focus on coffee consumption and MetS in patients with RA.

## 2. Materials and Methods

### 2.1. Study Population

The NHANES is a nationally representative cross-sectional study with a two-year cycle conducted by the National Center for Health Statistics (NCHS) of the Centers for Disease Control and Prevention (CDC) from 1971. It enrolls participants using a multistage probability and oversampling design, enabling a weighted analysis that reflects the health and nutritional status of the non-institutional US population. Initially, participants were interviewed at home, after which they were examined at a mobile examination center (MEC). During the interview, demographic, socioeconomic, health, and dietary data were collected. The dietary interview was conducted by the US Department of Agriculture (USDA) in partnership with the US Department of Health and Human Services using a general questionnaire format and two 24-h recalls obtaining detailed dietary intake information from NHANES participants. Initially, the first dietary recall interview was conducted in person at the MEC, followed by a telephone interview 3 to 10 days later. Interviewers who had undergone a week of rigorous training collected information on all foods and beverages consumed within the previous 24 h using the USDA Automated Multiple Pass Method. Laboratory tests, examinations, and questionnaire administration were also performed in a subset of participants.

In this study, secondary data analysis was conducted using eight consecutive cycles of the NHANES from 2003 to 2018. Patients with RA were identified according to the question “doctor ever said you had arthritis?” in the medical condition section of the interview. We excluded patients younger than 20 years of age and those with incomplete MetS z-score data. Ultimately, this study included 1094 eligible participants, with Appendix A showing the cohort construction flowchart.

### 2.2. Assessment of MetS Severity Based on Z-Score Data

Traditionally, the most widely used diagnostic criteria for MetS, including the National Cholesterol Education Program (NCEP/ATPIII) and International Diabetes Federation definitions, are based on setting cutoff values for each individual measure constituting MetS [13]. However, the traditional definition is still limited in its ability to identify the critical state of MetS (e.g., a fasting glucose level of 6.0 mmol/L when the cutoff is 6.1 mmol/L) and diagnose it in different ethnic groups. The MetS z-score is a widely used score that quantifies the value of MetS latent factor for an individual rather than simply providing a definition [14,15,16]. This score was derived from the NHANES, which includes a nationally representative sample of the US population and indicates the current status of an individual’s MetS in comparison to other people within the US [14]. The value of the MetS z-score represents the number of standard deviations from the mean of the US population. Therefore, with 0 as the cutoff, z-scores exceeding or below this threshold would indicate a greater and lower than average state of MetS, respectively. This approach allows the assessment of the degree of MetS, regardless of its presence or absence. The calculate formulas of MetS BMI z-score for male and female are presented in Figure 1 and Figure 2, respectively. 

Based on the self-reported demographic questionnaires, race/ethnicity was divided into four categories, namely non-Hispanic whites, non-Hispanic blacks, Mexican Americans, and non-Hispanic multiracial. We excluded patients from the non-Hispanic multiracial group due to the scoring criteria. Body mass index (BMI) was examined at the MEC. Laboratory tests, including fasting glucose, HDL, and triglycerides, were collected to calculate the z-score. Finally, the MetS z-scores of the corresponding participants were obtained according to a formula described elsewhere [14].

### 2.3. Assessment of Coffee, Decaffeinated Coffee and Caffeine Intake

Based on the two 24-h dietary recall interviews, data on dietary intake was extracted. Caffeine consumption from any source reported by participants and the coffee (caffeine contained) and decaffeinated coffee consumption were recorded based on USDA food codes. We converted the amount of coffee and decaffeinated coffee consumed into 6-ounce servings and categorized them by number of cups (nondrinkers, 0–2 cups per day, or >2 cups per day) [17]. Patients were also divided into three groups according to daily caffeine consumption (<100, 100–200, and >200 mg/day) [17].

### 2.4. Covariates

We investigated some potential confounders: age, education level, poverty–income ratio (PIR), low-density lipoprotein (LDL), insulin, waist circumference, use of glucocorticoids, alcohol, smoking exposure, and total energy intake. Education level and family income were included as socioeconomic factors. Patients were categorized into the following groups according to education level: <9th grade, 9–11th grade, high school graduate, some college, and college or above. Family income was reflected by the poverty–income ratio. Alcohol consumption status was classified based on whether it exceeded four drinks per day in the past 12 months [18]. Those who had smoked at least 100 cigarettes in their lifetime were determined as having smoking exposure [19]. Insulin levels and the Homeostatic Model Assessment for insulin resistance (HOMA-IR) were used to assess the degree of insulin resistance. Waist circumference and LDL, important factors reflecting the level of metabolic health, were also considered as covariates [13,20]. Glucocorticoid use was determined from the questionnaire section on prescription drugs. Because race/ethnicity, sex, BMI, triglycerides, fasting glucose, systolic blood pressure, and HDL are part of the z-score formula, so they were not used as covariates when the z-score was used as the dependent variable. Coffee is composed of caffeine and other compounds, and to clarify the effect of coffee on the metabolic syndrome, we selected decaffeinated coffee and caffeine intake as control groups. In the two control groups, we conducted the same analysis and introduced the same covariates.

### 2.5. Statistical Analysis

Statistical analysis was conducted using STATA (version 17.0) and R version 3.4.3, with two-tailed *p* < 0.05 indicating statistical significance. In accordance with the NHANES analytical guidelines, adjusted sample weights were constructed to address the bias associated with sample selection, oversampling, and nonresponse. Categorical variables were presented as percentages and 95% confidence intervals (95%CIs), whereas continuous variables were expressed as means ± standard errors. Categorical variables were compared using the chi-square test. Comparisons between groups were conducted using weighted linear regression. We used binary logistic regression to evaluate the effect of coffee on the prevalence of MetS (with or without MetS).

We used decaffeinated coffee and total caffeine consumption as controls, and four models were developed using linear regression to determine the association between coffee consumption, decaffeinated coffee consumption, caffeine, and MetS z-score. Model 1 did not adjust for any variable; model 2 adjusted for age, education level, and PIR; and model 3 adjusted for total energy intake, LDL, insulin, waist circumference, smoke exposure, drinking status, and use of glucocorticoid in addition to those adjusted in model 2. In addition, given that caffeine intake and decaffeinated coffee intake tended to have similar z-scores as coffee intake and the previous research showed the benefits of caffeine for MetS [21], we conducted a sensitivity analysis in which coffee intake, decaffeinated coffee intake, and total caffeine intake were used as covariates in the same model (model 4 and subgroup analysis). As components of the z-score, triglycerides, HDL, fasting glucose, and BMI were then analyzed as dependent variables using linear regression. Given the characteristics of the MetS z-score, we designated patients with a MetS z-score of <0 and ≥0 as the low- and high-risk groups, respectively. We then performed subgroup analysis in the two groups.

## 3. Results

### 3.1. Demographic Information and Clinical Features

From 2003–2018, a total of 1094 patients with RA were include in this study. The characteristics of the study sample grouped by coffee intake is described in Table 1. Participants were aged 60.87 ± 0.49 years, two in three were women, one in five were college educated, and three in five were white. Only 0.4% of the participants had used glucocorticoids in the past 1 month, whereas 6% consumed no more than four drinks of alcohol in the past 12 months. Among these patients, the weighted mean SBP, HDL, fasting glucose, and triglyceride was 127.24 ± 0.73 mmHg, 55.16 ± 0.61, 109.04 ± 1.02, and 141.48 ± 3.84 mg/dL, respectively. The weighted mean MetS z-score was 0.41 ± 0.04, with 66% of the patients scoring above 0. Among the included participants, significant differences in BMI (*p* = 0.03), MetS z-score (*p* = 0.03), SBP (*p* = 0.003), sex (*p* = 0.001), race (*p* < 0.001), drink status (*p* = 0.01), use of glucocorticoids (*p* = 0.003), and smoke exposure (*p* < 0.001) were observed between the different coffee consumption groups.

Appendix A compare the cohort characteristics according to decaffeinated coffee consumption and total caffeine intake, respectively. Overall, older people tended to drink more decaffeinated coffee (*p* < 0.001) and have lesser total caffeine intake (*p* < 0.001). Intake of decaffeinated coffee was associated with low total energy intake and high serum HDL compared to nonconsumption. However, more caffeine intake was found to be associated with low HDL levels and more total energy intake (compared to <100 mg; 100–200 mg/d: β-coefficient, 139.9, *p* = 0.001; >200 mg/d: β-coefficient, 283.5, *p* < 0.001).

### 3.2. Association between MetS Z-Score and Coffee, Decaffeinated Coffee, and Caffeine Intake

Three models were established to explore the association between MetS z-score and coffee, decaffeinated coffee, and caffeine intake (Table 2). We first assessed the relationship between coffee, decaffeinated coffee, and caffeine consumption on MetS z-score in model 1 with unadjusted variables. Only consumption of over two cups coffee per day was associated with a decrease in the MetS z-score (*p* = 0.03). After adjusting for age, PIR, and education level in model 2, we found that patients with a coffee consumption of >2 cups had lower MetS z-scores (*p* = 0.04) compared to nondrinkers. After considering additional variables mentioned in the Methods section in model 3, results similar to model 1 were observed, such that drinking over two cups coffee per day was consistently associated with a decrease in the MetS z-score. Details in model 3 were shown in Appendix A. In all three groups, several factors were correlated with an increase in the MetS z-score: increased waist circumference, high level of insulin, and high daily energy intake. Conversely, high PIR was associated with lower z-scores.

To investigate the reasons for the effect of coffee intake on z-score, we analyzed the relationship between the components of z-score and coffee. In Table 3, we performed a weighted linear regression with coffee intake as the independent variable. A β-coefficient was calculated by weighted linear regression for BMI, HDL, fasting glucose, triglyceride and SBP, respectively. After fully adjustment, we found that coffee intake was associated with lower BMI (*p* = 0.03 for 0–2 cups per day; *p* = 0.02 for >2 cups per day) and systolic blood pressure (*p* = 0.44 for 0–2 cups per day; *p* = 0.02 for >2 cups per day) compared to non-drinkers.

### 3.3. Association between Prevalence of MetS and Coffee Intake

We also divided participated into MetS group (169 participants) and non-MetS group (925 participants) based on the National Cholesterol Education Program (NCEP/ATPIII). Then, we used binary logistic regression to evaluate the effect of coffee on the prevelance of MetS (Appendix A). Both zero to two cups per day and more than two cups per day of coffee were not significantly associated with the development of MetS (OR: 1.07, *p* = 0.82; OR: 1.03, *p* = 0.91) after ful adjustment compared to non-drinkers.

### 3.4. Subgroup Analysis

Weighted linear regression with multiple variable adjustment was performed in both groups. In the high-risk group, high PIR, total energy intake, and waist circumference were associated with a lower MetS z-score (Table 4). Daily coffee intake, decaffeinated coffee intake, and caffeine intake were all correlated with a lower MetS z-score in the three control groups; however, significant findings were only observed among those who consumed >2 cups of coffee per day group (*p* = 0.03). Results of linear regression in the low-risk group are shown in Table 5. For participants with a MetS z-score of <0, daily intake of <2 cups of coffee was associated with low z-scores. However, no significant findings were observed among participants with daily coffee intake over two cups. Appendix A presents the association between coffee consumption and z-score components in the two groups. Participants in the high-risk group who consumed >2 cups of coffee per day had a lower BMI (*p* = 0.048) compared to nondrinkers. However, in the high-risk group, participants with a daily coffee intake of over two cups had higher fasting glucose levels (*p* = 0.03) than nondrinkers.

### 3.5. Sensitivity Analysis

We conducted sensitivity analysis by adding caffeine and decaffeinated coffee as independent variables based on model 3 (Appendix A). Accordingly, we found that only consuming >2 cups of coffee remained significantly associated with the MetS z-score (β, −0.20; 95% CI, −0.39 to −0.01; *p* = 0.04). All other stratified drink categories were not significantly associated with the MetS z-score. Furthermore, increased daily energy intake and waist circumference were associated with the severity of metabolic disorder, whereas PIR was a protective factor.

After repeating the above steps in the low-risk group, we found that daily coffee intake of <2 cups (β, −0.25; 95%CI, −0.39 to −0.01; *p* = 0.04) and daily caffeine intake >200 mg were significantly associated with the MetS z-score. However, decaffeinated coffee was not significantly associated with the MetS z-score regardless of the intake amount.

The same sensitivity analysis in the high-risk group showed that coffee consumption remained associated with a higher MetS z-score after adjusting for decaffeinated coffee and caffeine consumption, albeit with decreased statistical significance: >2 cups with β of −0.24 (95% CI, −0.49–0.00; *p* = 0.048).

## 4. Discussion

Coffee has been associated with metabolism outcomes among individuals with, or at risk for, MetS. These associations, however, remain controversial due to different study designs and study populations. Moreover, few studies have focused on the relationship between MetS and coffee in patients with RA. The current study therefore clarifies several aspects of coffee’s relationship to MetS status in patients with RA based on the NHANES 2003–2018 database. First, we found that coffee consumption was associated with the severity of MetS in patients with RA. Second, given that our data comes from a 16-year nationwide population survey, it is nationally representative and uniquely free of selection bias. Third, two 24-h dietary recalls (the gold standard for nutritional epidemiology) and MetS z-score (a widely used tool to evaluate MetS) were used to quantify coffee consumption and the severity of MetS, respectively. Finally, to address potential confounders, we introduced decaffeinated coffee and total caffeine intake as control groups and performed sensitivity analyses.

The findings of the current study showed that BMI was lower among those who consumed >2 cups of coffee per day, although no significant difference in total daily energy intake was observed. In fact, coffee has been shown to be beneficial for weight loss for over a decade [22]. This may be related to the biological function of caffeine. Caffeine is one of the important components of coffee, which belongs to a group of compounds called methylxanthines. The thermogenic effects of caffeine had been observed a few years ago. Although the mechanism has yet to be fully clarified, it is mainly thought to be due to the inhibitory effects of the phosphodiesterase-induced degradation of intracellular cyclic AMP (cAMP) and antagonism of adenosine receptors [23]. Kevin et al. also revealed the lipolytic effects of caffeine mediated via the sympathetic nervous system [24], although our study suggested no effects on lipid levels in patients with RA. Furthermore, caffeine increases energy expenditure while decreasing energy intake, thereby affecting the energy balance [22]. Thus, caffeine had been proposed as a strategy for weight loss and weight maintenance given its effects on thermogenesis, fat oxidation, and negative energy balance [22,25]. The above mechanism explains the association of coffee intake with lower BMI in our study. Interestingly, no significant correlation was observed between daily total caffeine intake and severity of MetS in our study. This may be attributed to two aspects. On the one hand, considering decaffeinated coffee consumption was also not associated with z-score, the effect of coffee on metabolic syndrome severity may be a combination of caffeine and other components such as phenolic compounds. On the other hand, we found that caffeine consumption was positively correlated, and this may due to the fact that other caffeine-rich foods, such as sweetened beverages, hot cocoa, and chocolate, are often associated with high fat and sugar, which may increase the risk of metabolic syndrome.

The present study found a negative correlation between coffee intake and blood pressure. Although caffeine intake causes a sharp increase in blood pressure, habitual coffee drinkers did not exhibit greater blood pressure with elevated levels of caffeine in the blood compared to non-habitual drinkers, which may due to other substances in coffee [26]. A large prospective cohort study that included 8780 participants without hypertension suggested an inverse relationship between moderate coffee intake (1–3 cups/day) and the risk of developing hypertension [27]. In addition, a dose–response meta-analysis of prospective cohort studies, which focused on the risk of relationship between increased coffee consumption and the risk of hypertension, also showed an inverse association [28]. The phenolic compounds contained in coffee have been widely believed to improve blood pressure [29]. Among them, chlorogenic acid and caffeic acid were proven to exert antihypertensive effects via their antioxidant properties [30]. Apart from the traditional risk factors for hypertension, other factors in RA may affect blood pressure and its control, such as inflammation, medications, and physical inactivity [31]. Elevated *C*-reaction protein (CRP), which reflects systemic inflammation, was found to reduce endothelial nitric oxide production, leading to vasoconstriction [32]. Additionally, CRP can upregulate angiotensin type-1 receptor expression, affect the renin–angiotensin system, and contribute to high blood pressure [33]. However, a metabolite of chlorogenic acid, ferulic acid, has been shown to repair endothelial function by increasing the bioavailability of nitric oxide [34]. Moreover, proinflammatory cytokines expressed by vascular endothelial cells were reduced in response to coffee polyphenols [8]. In summary, coffee can affect the blood pressure of individuals with RA through several ways.

However, the metabolic effects of coffee in RA are more complex. Unlike the normal population, the autoinflammation caused by immune dysfunction makes insulin resistance and MetS more prevalent and severe in RA, playing a key role in metabolic disorders [35]. A previous study by Dessein et al. found an association between insulin resistance and elevated markers of inflammation in RA [36]. Their study also showed a negative correlation between beta-cell function and disease activity. It can also be argued that the severity of insulin resistance and MetS reflect, to some extent, the state of immune disorders in rheumatoid patients. By reducing proinflammatory cytokines, macrophage and natural killer cell activity, and B and T cell proliferation, caffeine affects the immune system [37]. It is this immunomodulatory function that reduces the inflammation-related resistance in patients with RA. Therefore, our study showed the immunomodulatory effects of caffeine in the high-risk group might gradually negate its effects on insulin resistance and sensitivity with increasing intake (i.e., more than two cups of coffee per day), thereby reducing the risk of MetS while leaving blood glucose unaffected. In the low-risk group, who may have less inflammatory involvement, <2 cups of coffee per day already provided the appropriate immunomodulatory function; hence, their z-score was reduced. However, caffeine had a more pronounced effect on glucose and exceeded the immunomodulatory effect in those who consumed >2 cups of coffee per day; hence, these participants showed no decrease in z-scores and had higher glucose levels.

Dyslipidemia is also an important component of MetS. The effects of coffee on patients with RA are complex. On the one hand, the inflammation and treatment of RA itself can lead to changes in blood lipid levels. Several studies have shown a decrease in total, LDL and HDL cholesterol and triglycerides in RA with high-grade inflammation; however, no such decrease was observed in stable disease, non-inflammatory arthritis, or normal controls [38,39]. This could be attributed to inflammation-activated oxidation or scavenger receptor pathways, leading to a negative balance in cholesterol synthesis, or to the production of autoantibodies against LDL [40]. Regarding therapeutic drugs, hydroxychloroquine has been proven beneficial for lipid profiles; however, glucocorticoid use contributes to dyslipidemia [41]. Nonetheless, most investigators agree that coffee intake is associated with an increased risk for dyslipidemia [42]. The pooled results of a recent meta-analysis that included 12 RCT demonstrated a significant positive association between coffee and total cholesterol, triglycerides, and LDL cholesterol [42]. Evidence has shown that either the lipid compounds contained in coffee or individual genetic variation may be responsible for the reduced synthesis and release of bile acids [42]. However, chlorogenic acid, the main phenolic compound in coffee, has been found to have anti-lipid properties [43]. Therefore, the mechanism by which coffee exerts its lipid metabolizing effects remains unclear. The current study found that coffee, decaffeinated coffee, and caffeine intake had no effect on lipid profiles in patients with RA, which may suggest that coffee and inflammation act together and then achieve a balance. More research is needed to determine the mechanism by which coffee affects lipids in patients with RA.

Interestingly, when patients were divided into groups with and without metabolic syndrome, we found no significant association between coffee and the development of metabolic syndrome. Actually, lots of studied have focused on the relationship between coffee and prevalence of MetS, whereas the results are controversial [9,10,11,12]. This may be due to selection bias, study design, and differences sample size. In our study, for example, the difference in the sample size between the two groups (with or without MetS) was significant, which may cause statistical errors. In addition, there are differences in the five widely used classification criteria for MetS, which could also lead to selection bias. Therefore, we mainly used the MetS z-score with higher sensitivity in this study to help us define whether patients have metabolic disorders, which is also meaningful in practical clinical applications to achieve early diagnosis and intervention. In fact, our findings do suggest that the z-score can identify more people at risk for metabolic disorders (730 for z-score vs. 169 for NCEP/ATPIII). However, follow-up studies are needed to compare the efficacy of the z-score and the classification criteria.

The present study has several strengths. Primarily, the data for our study came from the NHANES, thereby ensuring high quality. Moreover, we included a wide range of potential confounders closely associated with RA and MetS to provide more reliable results. Moreover, we used decaffeinated coffee and caffeine intake as control groups and conducted sensitivity analyses to increase the plausibility of our results.

Nonetheless, our study has several limitations worth noting. First, this was a cross-section study, which limits causal inferences. In addition, we lacked data on inflammatory markers, which are important for patients with RA, although these autoinflammatory conditions ultimately affect MetS by influencing its components. Moreover, due to missing data, we could not categorize coffee according to bean type, roasting, and brewing method. Last, although our application of the MetS z-score to determine a patient’s risk of MetS may not be the most comprehensive, and some studies have shown this scoring system to be reliable [15,16].

## 5. Conclusions

Our study suggested that daily intake of coffee was associated with low severity of MetS in patients with self-reported RA. Drinking no more than zero to two cups and over two cups of coffee was inversely associated with the severity of MetS among those whose z-scores do not exceed 0 and more than 0, respectively. This may be due to the beneficial effects of coffee on BMI, blood pressure, and insulin resistance. However, no significant association was observed between decaffeinated coffee consumption and daily caffeine intake and the severity of MetS. Further large-scale prospective studies are needed to examine the relationship between coffee intake and risk of MetS in RA.

## Figures and Tables

**Figure 1 nutrients-15-00107-f001:**
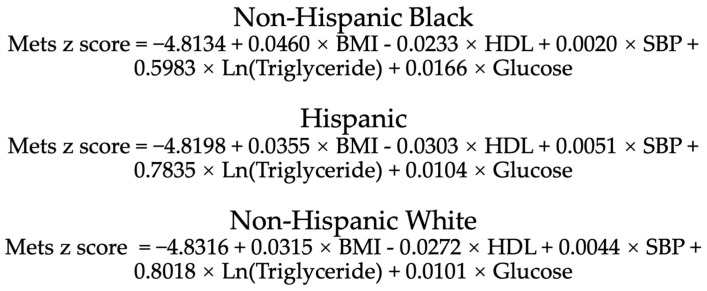
Calculation formula of MetS BMI z-score for male of different races. BMI, body mass index; HDL, high density lipoprotein; SBP, systolic blood pressure. BMI: kg/m^2^; HDL, triglyceride, glucose: mg/dl; SBP: mmHg.

**Figure 2 nutrients-15-00107-f002:**
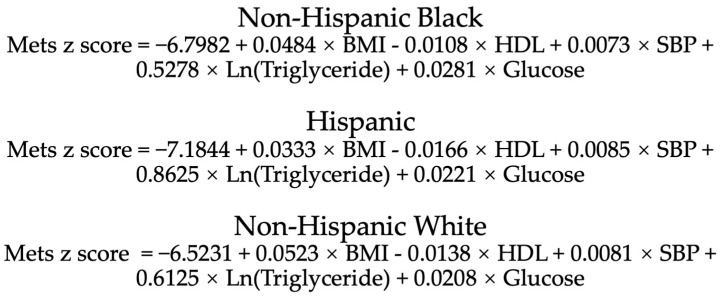
Calculation formula of MetS BMI z-score for female of different races. BMI, body mass index; HDL, high density lipoprotein; SBP, systolic blood pressure. BMI: kg/m^2^; HDL, triglyceride, glucose: mg/dl; SBP: mmHg.

**Table 1 nutrients-15-00107-t001:** Characteristics of the study population grouped by coffee intake, National Health and Nutrition Examination Survey (NHANES) 2003–2018 (n = 1094).

Characters	Total (*n* = 1094)	Nondrinker (*n* = 457)	0–2 Cups/day (*n* = 283)	>2 Cups/day (*n* = 354)	*p* Value
Age	60.87 ± 0.49	61.59 ± 0.79	61.63 ± 1.06	59.67 ± 0.78	0.065
BMI (kg/m^2^)	30.2 ± 0.26	30.91 ± 0.46	29.83 ± 0.49	29.7 ± 0.41	0.03
PIR	2.92 ± 0.06	2.87 ± 0.09	2.81 ± 0.13	3.02 ± 0.09	0.16
Energy (kcal)	1952.68 ± 29.43	1904.89 ± 41.34	1918.5 ± 63.08	2022.41 ± 51.92	0.06
MetS Z-score	0.41 ± 0.04	0.51 ± 0.06	0.41 ± 0.07	0.32 ± 0.06	0.026
>0	66.73 (0.31–0.36)	68.1 (0.64–0.72)	64.7 (0.59–0.70)	66.7 (0.62–0.71)	
<0	33.27 (0.64–0.67)	31.95 (0.28–0.36)	35.3 (0.30–0.41)	33.3 (0.29–0.38)	
SBP (mmHg)	127.24 ± 0.73	129.31 ± 1.12	128.01 ± 1.52	124.66 ± 1.2	0.003
Insulin (μU/mL)	13.49 ± 0.54	14.36 ± 1.03	13.4 ± 1.06	12.65 ± 0.72	0.2
Total cholesterol (mg/dL)	5.15 ± 0.04	5.1 ± 0.07	5.17 ± 0.09	5.18 ± 0.07	0.53
HDL (mg/dL)	55.16 ± 0.61	55.7 ± 1.04	53.91 ± 1.13	55.36 ± 0.97	0.35
Fasting glucose (mg/dL)	109.04 ± 1.02	111.06 ± 1.96	108.73 ± 1.86	107.17 ± 1.39	0.19
Triglyceride (mg/dL)	141.48 ± 3.84	137.73 ± 4.48	141.37 ± 7.47	145.37 ± 7.76	0.52
HOMA-IR	71.83 ± 4.56	80.1 ± 10.02	69.94 ± 6.85	64.54 ± 4.65	0.15
Sex					0.001
Male	39 (0.36–0.42)	31.1 (0.26–0.37)	39.6 (0.32–0.48)	43 (0.37–0.49)	
Female	61 (0.58–0.64)	68.9 (0.63–0.74)	60.4 (0.52–0.68)	57 (0.51–0.63)	
MetS					0.75
With MetS	15.1 (0.13–0.18)	14.1 (0.11–0.19)	15.8 (0.11–0.23)	15.7 (0.12–0.21)	
Without MetS	84.9 (0.82–0.87)	85.9 (0.81–0.89)	84.2 (0.77–0.89)	84.3 (0.79–0.88)	
Race					<0.001
Mexican American	10.15 (0.08–0.12)	8.75 (0.06–0.12)	12.01 (0.09–0.16)	10.45 (0.08–0.14)	
Other Hispanic	7.13 (0.06–0.09)	5.03 (0.03–0.07)	13.07 (0.1–0.18)	5.08 (0.03–0.08)	
Non-Hispanic White	61.33 (0.58–0.64)	55.36 (0.51–0.6)	52.65 (0.47–0.58)	75.99 (0.71–0.8)	
Non-Hispanic Black	21.39 (0.19–0.24)	30.85 (0.27–0.35)	22.26 (0.18–0.27)	8.47 (0.06–0.12)	
Education					0.44
Less Than 9th Grade	11.06 (0.09–0.13)	10.5 (0.08–0.14)	14.13 (0.11–0.19)	9.32 (0.07–0.13)	
9–11th Grade	15.27 (0.13–0.18)	15.1 (0.12–0.19)	17.31 (0.13–0.22)	13.84 (0.11–0.18)	
High School	24.59 (0.22–0.27)	26.26 (0.22–0.3)	21.91 (0.17–0.27)	24.58 (0.2–0.29)	
Some College	28.43 (0.26–0.31)	27.79 (0.24–0.32)	27.21 (0.22–0.33)	30.23 (0.26–0.35)	
College Graduate	20.66 (0.18–0.23)	20.35 (0.17–0.24)	19.43 (0.15–0.24)	22.03 (0.18–0.27)	
Alcoholic > 4 drinks					0.01
Yes	78 (0.75–0.81)	79 (0.74–0.83)	78 (0.70–0.84)	78 (0.72–0.83)	
No	6 (0.04–0.08)	4 (0.02–0.07)	4 (0.02–0.08)	9 (0.06–0.14)	
Not recorded	16 (0.13–0.19)	17 (0.14–0.22)	18 (0.12–0.25)	13 (0.10–0.18)	
Glucocorticoid					0.003
YES	0.4 (0.00–0.02)	0.4 (0.00–0.03)	-	1 (0.00–0.04)	
No	97.6 (0.96–0.99)	98.8 (0.97–1.00)	99 (0.98–1.00)	95 (0.92–0.97)	
Not recorded	2 (0.01–0.03)	0.8 (0.00–0.02)	1 (0.00–0.02)	4 (0.02–0.07)	
Smoke					<0.001
Yes	47.26 (0.44–0.5)	45.73 (0.41–0.5)	43.82 (0.38–0.5)	68.36 (0.63–0.73)	
No	52.56 (0.5–0.56)	54.27 (0.5–0.59)	55.48 (0.5–0.61)	31.64 (0.27–0.37)	
Not recorded	0.18 (0–0.01)	-	0.71 (0–0.03)	-	

Data are presented as means (SE) for continuous measures, and as percentage (95% confidence interval) for categoric measures.; *p* value was calculated by weighted linear regression model for continuous variables and chi-square test for categorical variables, respectively. BMI, body mass index; PIR, poverty-income ratio; SBP, systolic blood pressure; HDL, high density lipoprotein; SE, standard error; CI, confidence interval.

**Table 2 nutrients-15-00107-t002:** Weighted linear regression for MetS z-score in coffee, decaffeinated coffee, and caffeine intake group.

Characteristic	Model 1 *		Model 2 **		Model 3 ***	
	Coefficient (95% CI)	*p*	Coefficient (95% CI)	*p*	Coefficient (95% CI)	*p*
Coffee (ref. = nondrinkers)						
0–2 cups	−0.09 (−0.28–0.09)	0.3	−0.10 (−0.29–0.08)	0.29	−0.07 (−0.22–0.07)	0.33
>2 cups	−0.19 (−0.35 to −0.02)	0.03	−0.17 (−0.33 to −0.01)	0.04	−0.13 (−0.24 to −0.01)	0.04
Decaffeinated Coffee (ref. = nondrinkers)						
0–2 cups	−0.07 (−0.30–0.15)	0.52	−0.07 (−0.30–0.16)	0.53	−0.01 (−0.19–0.16)	0.89
>2 cups	−0.05 (−0.35–0.25)	0.75	−0.05 (−0.34–0.23)	0.73	−0.15 (−0.36–0.05)	0.15
Caffeine intake (ref. = 0–100 mg)						
100–200 mg	0.08 (−0.10–0.25)	0.40	0.12 (−0.06–0.30)	0.2	0.02 (−0.12–0.15)	0.79
>200 mg	−0.07 (−0.24–0.10)	0.41	−0.02 (−0.19–0.15)	0.8	−0.06 (−0.19–0.06)	0.31

* Model 1 did not adjust for any other variables. ** Model 2 was adjusted for age, education level and PIR. *** Model 3 was adjusted for age, education level, PIR, total energy intake, LDL, fasting insulin, waist circumference, smoke and alcohol exposure, and use of glucocorticoid. CI, confidence interval; Ref., reference. A β-coefficient was calculated by weighted linear regression for MetS z-score.

**Table 3 nutrients-15-00107-t003:** Weighted linear regression for the components of the z-score in coffee intake group.

Components	0–2 Cups/day (Ref. = Nondrinker)		>2 Cups/day (Ref. = Nondrinker)	
	Coefficient (95% CI)	*p*	Coefficient (95% CI)	*p*
BMI (kg/m^2^) *	−0.7 (−1.37 to −0.09)	0.03	−0.7 (−1.30 to −0.10)	0.02
HDL (mg/dL)	−1.80 (−4.81–1.22)	0.24	−0.34 (−3.14–2.46)	0.81
Fasting glucose (mg/dL)	−0.13 (−0.42–0.17)	0.4	−0.22 (−0.48–0.05)	0.11
Triglyceride (mg/dL)	3.65 (−13.47–20.76)	0.68	7.64 (−9.96–25.25)	0.39
SBP (mmHg)	−1.38 (−4.89 −2.14)	0.44	−3.58 (−6.69 to −0.46)	0.02

* Adjusted for age, gender, education level, total energy intake, PIR, HOMA-IR, smoke status, triglyceride, total cholesterol, waist circumference and use of glucocorticoid. BMI, body mass index; HDL, high density lipoprotein; SBP, systolic blood pressure; CI, confidence interval; Ref., reference. A β-coefficient was calculated by weighted linear regression for BMI, HDL, fasting glucose, triglyceride and SBP, respectively.

**Table 4 nutrients-15-00107-t004:** Adjusted association between coffee, decaffeinated coffee, and caffeine intake and MetS z-score in high-risk group.

Characteristic	High-Risk Group (*n* = 730)					
	Coffee		Decaffeinated Coffee		Caffeine	
	Coefficient (95% CI)	*p*	Coefficient (95% CI)	*p*	Coefficient (95% CI)	*p*
Age	0.00 (−0.01–0.00)	0.44	0.00 (−0.01–0.00)	0.56	0.00 (−0.01–0.00)	0.35
Education (ref. = Less Than 9th Grade)						
9–11th Grade	−0.12 (−0.48–0.23)	0.5	−0.13 (−0.49–0.22)	0.46	−0.11 (−0.47–0.24)	0.53
High School	−0.20 (−0.53–0.12)	0.23	−0.22 (−0.55–0.10)	0.18	−0.20 (−0.53–0.12)	0.21
Some College	−0.22 (−0.56–0.11)	0.19	−0.24 (−0.58–0.10)	0.17	−0.22 (−0.55–0.11)	0.2
College	−0.19 (−0.53–0.15)	0.28	−0.20 (−0.54–0.15)	0.26	−0.18 (−0.52–0.16)	0.29
PIR	−0.08 (−0.13 to −0.03)	0.002	−0.08 (−0.13 to −0.03)	<0.001	−0.08 (−0.13 to −0.03)	0.002
Energy (kcal)	0.00 (0.00–0.00)	0.25	0.00 (0.00–0.00)	0.19	0.00 (0.00–0.00)	0.25
LDL (mg/dL)	−0.03 (−0.10–0.04)	0.43	−0.03 (−0.10–0.04)	0.41	−0.03 (−0.10–0.04)	0.42
Insulin (μU/mL)	0.01 (0.00–0.02)	0.02	0.01 (0.00–0.02)	0.01	0.01 (0.00–0.02)	0.01
Waist Circumference (cm)	0.02 (0.01–0.02)	<0.001	0.02 (0.01–0.02)	<0.001	0.02 (0.01–0.02)	<0.001
Smoke (ref. = Yes)						
No	−0.07 (−0.19–0.05)	0.26	−0.03 (−0.16–0.09)	0.6	−0.05 (−0.17–0.08)	0.45
Not recorded	-	-	-	-	-	-
Alcoholic > 4 drinks (ref. = Yes)						
No	−0.20 (−0.49–0.09)	0.97	−0.21 (−0.49–0.08)	0.16	−0.20 (−0.48–0.08)	0.16
Not recorded	0.00 (−0.18–0.19)	0.18	0.01 (−0.17–0.20)	0.89	0.01 (−0.16–0.19)	0.88
Glucocorticoid (ref. = No)						
YES	0.24 (−0.27–0.75)	0.36	0.20 (−0.23–0.62)	0.36	0.21 (−0.25–0.66)	0.37
Not recorded	−0.30 (−0.55 to −0.06)	0.01	−0.36 (−0.59 to −0.12)	<0.001	−0.34 (−0.59 to −0.09)	0.01
Coffee (ref. = nondrinkers)						
0–2 cups	−0.04 (−0.21–0.13)	0.66	-	-	-	-
>2 cups	−0.16 (−0.31 to −0.01)	0.03	-	-	-	-
Decaffeinated Coffee (ref. = nondrinkers)						
0–2 cups	-	-	−0.09 (−0.28–0.09)	0.31	-	-
>2 cups	-	-	−0.04 (−0.27–0.19)	0.73	-	-
Caffeine intake (ref. = 0–100 mg)						
100–200 mg	-	-	-	-	0.01 (−0.15–0.17)	0.89
>200 mg	-	-	-	-	−0.08 (−0.23–0.06)	0.24

Adjusted model included age, education level, PIR, total energy intake, LDL, fasting insulin, waist circumference, smoke and alcohol exposure, use of glucocorticoid. BMI, body mass index; HDL, high density lipoprotein; CI, confidence interval; Ref., reference. A β-coefficient was calculated by weighted linear regression for MetS z-score.

**Table 5 nutrients-15-00107-t005:** Adjusted association between coffee, decaffeinated coffee, and caffeine intake and MetS z-score in low-risk group.

Characteristic	Low-Risk Group (*n* = 364)					
	Coffee		Decaffeinated Coffee		Caffeine	
	Coefficient (95% CI)	*p*	Coefficient (95% CI)	*p*	Coefficient (95% CI)	*p*
Age	0.00 (0.00–0.00)	0.82	0.00 (0.00–0.00)	0.92	0.00 (0.00–0.00)	0.55
Education (ref.= Less Than 9th Grade)						
9–11th Grade	−0.05 (−0.23–0.12)	0.54	−0.05 (−0.22–0.11)	0.53	−0.08 (−0.25–0.10)	0.38
High School	−0.02 (−0.19–0.15)	0.83	−0.01 (−0.18–0.15)	0.89	−0.02 (−0.19–0.15)	0.83
Some College	−0.11 (−0.29–0.07)	0.22	−0.10 (−0.27–0.06)	0.22	−0.09 (−0.27–0.08)	0.3
College	−0.19 (−0.53–0.15)	0.53	−0.05 (−0.23–0.12)	0.54	−0.06 (−0.25–0.12)	0.49
PIR	−0.03 (−0.06–0.00)	0.09	−0.03 (−0.06–0.01)	0.15	−0.03 (−0.06–0.01)	0.14
Energy (kcal)	0.00 (0.00–0.00)	0.31	0.00 (0.00–0.00)	0.26	0.00 (0.00–0.00)	0.21
LDL (mg/dL)	0.07 (0.02–0.13)	0.01	0.07 (0.02–0.12)	0.01	0.07 (0.02–0.13)	0.01
Insulin (μU/mL)	0.02 (0.01–0.03)	<0.001	0.02 (0.01–0.03)	<0.001	0.02 (0.01–0.03)	<0.001
Waist Circumference (cm)	0.01 (0.01–0.02)	<0.001	0.01 (0.01–0.01)	<0.001	0.01 (0.01–0.01)	<0.001
Smoke (ref. = Yes)						
No	−0.02 (−0.12–0.07)	0.62	−0.02 (−0.12–0.08)	0.63	−0.05 (−0.14–0.05)	0.31
Not recorded	0.34 (0.05–0.63)	0.02	0.24 (−0.05–0.53)	0.1	0.27 (−0.03–0.57)	0.08
Alcoholic > 4 drinks (ref. = Yes)						
No	0.00 (−0.13–0.12)	0.15	−0.17 (−0.41–0.07)	0.17	−0.12 (−0.35–0.11)	0.31
Not recorded	−0.18 (−0.42–0.06)	0.96	0.00 (−0.13–0.12)	0.95	−0.01 (−0.13–0.12)	0.92
Glucocorticoid (ref. = No)						
YES	0.91 (0.73–1.10)	<0.001	−0.89 (−1.06-−0.72)	0.53	−0.81 (−1.00 to−0.61)	<0.001
Not recorded	0.97 (0.68–1.27)	<0.001	0.08 (−0.17–0.32)	<0.001	0.07 (−0.15–0.30)	0.54
Coffee (ref. =nondrinkers)						
0–2 cups	−0.12 (−0.23 to −0.01)	0.03			-	-
>2 cups	0.00 (−0.12–0.12)	0.97	-	-	-	-
Decaffeinated Coffee (ref. = nondrinkers)						
0–2 cups	-	-	−0.12 (−0.31–0.07)	0.22	-	-
>2 cups	-	-	−0.02 (−0.20–0.16)	0.81	-	-
Caffeine intake (ref. = 0–100 mg)						
100–200 mg	-	-	-	-	0.04 (−0.07–0.15)	0.45
>200 mg	-	-	-	-	−0.08 (−0.21–0.04)	0.19

Adjusted model included age, education level, PIR, total energy intake, LDL, fasting insulin, waist circumference, smoke and alcohol exposure, use of glucocorticoid. BMI, body mass index; HDL, high density lipoprotein; CI, confidence interval; Ref., reference. A β-coefficient was calculated by weighted linear regression for MetS z-score.

## Data Availability

The NHANES 2003–2018 data used in the current manuscript can be downloaded from CDC website: https://wwwn.cdc.gov/nchs/nhanes/Default.aspx, accessed on 25 December 2022.

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
