# Peer review of "Association between Coffee Consumption, Caffeine Intake, and Metabolic Syndrome Severity in Patients with Self-Reported Rheumatoid Arthritis: National Health and Nutrition Examination Survey 2003–2018"

_nutrients, 2022, doi:10.3390/nu15010107_

Round 1

Reviewer 1 Report

Wang et al. present an interesting study investigating the association between coffee consumption and metabolic syndrome in rheumatoid arthritis patients. This is a well-designed study and has interesting results, however the manuscript would benefit if results are more focused and presented concisely to avoid confusion and move some of the results to the supplementary data. Here are my comments and some suggestions for the authors:

1.     Authors have considered MetS Z-score as continuous variable to define the severity of the metabolic syndrome which seems to be appropriate approach. It would be worth exploring the associations by considering the categorical definition of MetS i.e., presence or absence of MetS.

2.     A brief description of MetS Z-score calculation should be provided. Supplementary tables 4 and 5 list only four components. Was systolic blood pressure not included in the calculation?

3.     It would be interesting to include the Supplementary Table 4 as a main table and a section discussing the relationship with the components of MetS could be included. It appears that only BMI seems to drive the association. It would be worth discussing that and if there are any previous reports.

4.  Table 2: For clarity, it would be good to present only variables that are significant in the main table. The table in the current form could be moved to the supplementary data. Supplementary Table 3 could be included as main table as it concisely presents the overall findings of the study.

5.  Page 3: “We converted the amount of coffee and decaffeinated coffee consumed into 6-ounce servings and categorized them by number of cups (nondrinkers, 0–2 cups per day, or >3 cups per day)” Is there a typo? Is the correct number >2 cups per day?

6.  Provide the number of individuals in each of the group- nondrinkers, 0-2 cups and so on.

7.  Page 4: “whereas 6% drank more than four drinks per day.” To avoid confusion, it would be good to mention “alcoholic drinks”.

8.  Page 4: “Using decaffeinated coffee and total caffeine consumption as controls,” this is not clear. please elaborate.

9.  Table 1: Check the percentages of males and females, they don’t add up to 100.

Author Response

we have uploaded it as a word file. Please see the attachment.

Reviewer 2 Report

Review of the work entitled

Association between coffee consumption, caffeine intake, and metabolic syndrome severity in patients with rheumatoid arthritis: National Health and Nutrition Examination Survey 2003–2018

Authors: Shuolin Wang , Yingdong Han , He Zhao , Xinxin Han , Yue Yin , Juan Wu , Yun Zhang * , Xuejun Zeng *

The study was conducted on the results obtained from a project with significant impact and credibility. However, two of its aspects, i.e. the status of the study group concerning the occurrence of RA and the overly extensive and redundant statistical analysis, require further attention

Firstly, it is doubtful to state categorically that the study was conducted among people suffering from rheumatoid arthritis. The description in the MM shows that patients were qualified based on the answer to the question of whether the doctor ever mentioned RA. This means that the group described by the authors as RA includes patients with an unknown disease status, patients with active disease or in remission, treated and untreated patients. This fact significantly reduces the credibility of the results obtained concerning this particular disease. Therefore, both the title of the paper and the conclusions should be drawn in relation to persons suspected of having RA currently or in the past. Secondly, the statistical analysis of the results requires supplementation and editing due to low readability for the recipient.

Details in this regard are presented below:

1)      A clear description of the grouping criteria is needed.

-          it is not entirely clear in Tab 1 whether decaffeinated coffee drinkers are included in the 0-2 and > 2 cups/day groups.

-          description of the tab. 1 is stated as “grouped by caffeine intake” when groups are divided by the amount of coffee (cups per day). It seems incorrect nomenclature regarding the 2.3 paragraph?

-          Are the caffeine intake also assessed for  decaffeinated coffee drinkers?

2)      Statistical analysis -covariates:

The authors indicated in section 2.4 that some of the variables will not be included in the tested models because that they are factors directly affecting the MetS z-score.In this situation, it is necessary to explain the legitimacy of including parameters such as total cholesterol, HDL, HOMA, glucose, insulin and waist circumference in the multivariate models presented in Tab. 2, 3, 4. MetS z-score takes into account these parameters, so the analysis should concern the z-score itself or people with identified MetS. In this context, the analyzes presented in Table 4 Supplementary Materials regarding BMI and HOMA in Table 5 are redundant. All the more so as none of these parameters are included in the Conclusions.

3)      Statistical analysis – sensitivity analysis

Although we understand the rationale for performing sensitivity analysis, which the authors put in section 2.5, its use seems to be redundant.

Mainly because it does not bring any new information concerning previously analysed models.

In general, there are too many tables in the work and additional materials. Their analysis leads to the same conclusion that the reader seems lost in this thicket.

It is necessary to present the results in a more synthetically and transparently way.

4)      Results

The main issue that should be disused and elucidated is the lack of relationship MetS z-score with the total caffeine intake in contrast to the number of coffee cups a day. Is it triggered by caffeine from other than coffee drinks sources? What is the possible reason for that situation?  

5)      Conclusion

The first two sentences of the conclusion chapter seem to be mutually exclusive. The reader may be confused as to which coffee consumption is associated with lower risk. The conclusions in their current form appear to be a summary of Table 1 for the first sentence and Table 4 for the second sentence, rather than an overall summary of all the results and discussions.

Others comments:

There is a mistake in spelling HOMA in description of Supplementary Table 5.

Author Response

(The authors gave the same response as above.)

Round 2

Reviewer 1 Report

Authors have adequately addressed all the concerns. I don't have any further comments.

Author Response

Thank you for the constructive comments.

Reviewer 2 Report

The authors significantly improved the quality of the manuscript and satisfactorily responded to all the reviewer's comments. In its current form, the work is readeable and consistent. 

However, I am forced to point out the inaccuracies in the new description of the statistical analysis. 

In paragraph 2.5, the authors added that they compared the groups using weighted regression analysis and LOGISIC REGRESSION.

Logistic regression is not used to compare groups, but to models a relationship between predictor variables and a categorical response variable. The results of the logistic analysis should be reported as an Odd Ratio with the appropriate confidence interval and probability. It is necessary for the authors to indicate in which table the results of the regression analysis were placed.

In Chapter 3.2, the authors indicate that the correlation results are presented in Table 3. The description of statistical methods does not mention correlation analysis (most often reported as correlation coefficient r with probability p). Since correlation analysis is not mentioned elsewhere in the manuscript, and the results in Table 3 are reported in the same way as for weighted linear regression, it is necessary to re-examine this paragraph.
